# Cyclophilin A Promotes Inflammation in Acute Kidney Injury but Not in Renal Fibrosis

**DOI:** 10.3390/ijms21103667

**Published:** 2020-05-22

**Authors:** Khai Gene Leong, Elyce Ozols, John Kanellis, David J. Nikolic-Paterson, Frank Y. Ma

**Affiliations:** 1Department of Nephrology, Monash Health, Monash Medical Centre, Clayton, Victoria 3168, Australia; khaigeneleong@gmail.com (K.G.L.); elyce.ozols@monash.edu (E.O.); john.kanellis@monash.edu (J.K.); frank.ma@monash.edu (F.Y.M.); 2Monash University Centre for Inflammatory Diseases, Monash Medical Centre, Clayton, Victoria 3168, Australia

**Keywords:** acute kidney injury, chronic kidney disease, cyclophilin A, fibrosis, inflammation, renal fibrosis, tubular necrosis

## Abstract

Cyclophilin A (CypA) is a highly abundant protein in the cytoplasm of most mammalian cells. Beyond its homeostatic role in protein folding, CypA is a Damage-Associated Molecular Pattern which can promote inflammation during tissue injury. However, the role of CypA in kidney disease is largely unknown. This study investigates the contribution of CypA in two different types of kidney injury: acute tubular necrosis and progressive interstitial fibrosis. *CypA (Ppia)* gene deficient and wild type (WT) littermate controls underwent bilateral renal ischaemia/reperfusion injury (IRI) and were killed 24 h later or underwent left unilateral ureteric obstruction (UUO) and were killed 7 days later. In the IRI model, *CypA^−/−^* mice showed substantial protection against the loss of renal function and from tubular cell damage and death. This was attributed to a significant reduction in neutrophil and macrophage infiltration since *CypA^−/−^* tubular cells were not protected from oxidant-induced cell death in vitro. In the UUO model, *CypA^−/−^* mice were not protected from leukocyte infiltration or renal interstitial fibrosis. In conclusion, CypA promotes inflammation and acute kidney injury in renal IRI, but does not contribute to inflammation or interstitial fibrosis in a model of progressive kidney fibrosis.

## 1. Introduction

Cyclophilins are ubiquitously expressed proteins which belong to the immunophilin family [1,2]. All cyclophilins possess the “peptidyl-prolyl isomerase” (PPIase) activity that catalyses the interconversion of *cis* and *trans* isomers of proline to facilitate protein folding [2,3]. Cyclophilin A (CypA) is a highly abundant cytoplasmic protein that is expressed by virtually all mammalian cells [1,2]. Beyond its homeostatic role, CypA can contribute to the inflammatory response. CypA can be released from cells via active secretion, or passively during necrotic cell death, and bind to CD147 on the surface of leukocytes, including neutrophils, monocyte/macrophages and T cells. In vitro studies have demonstrated that CypA can promote monocyte and neutrophil migration, and macrophage activation [4,5,6]. Indeed, *CypA* gene-deficient mice are protected from acetaminophen-induced liver toxicity and inflammation, leading to the description of CypA as a Damage-Associated Molecular Pattern [7]. Indeed, the administration of supraphysiologic doses of recombinant CypA to mice can induce systemic inflammation [8]. CD147, the only known CypA receptor, is also expressed by many non-leukocyte populations, including tubular epithelial cells of the kidney [1,9,10]. Furthermore, CD147 is a scavenger receptor which can bind many other ligands, including leukocyte integrins, Selectin E, CD44 and S100A9 [11]. Indeed, *Cd147* gene-deficient mice are sterile with a variety of abnormalities, consistent with CD147 being a receptor for multiple ligands [12,13].

Acute kidney injury (AKI) is clinically defined as an acute increase in serum creatinine (>27 mmol/L within 48 h or >1.5-fold over a week) or loss of urine output. AKI is commonly seen in the emergency department where a variety of pre-renal causes (e.g., severe blood loss, major cardiac or abdominal surgery, sepsis, severe dehydration) result in low blood pressure and hypo-perfusion of the kidney [14,15]. In addition, acute kidney injury can result from acute tubular necrosis induced by nephrotoxic agents, including chemotherapeutic drugs, environmental toxins, contrast media and drug overdose [16]. Severe AKI is associated with high mortality rates and necessitates immediate dialysis [14,17], while those recovering from AKI are at increased risk of developing, or exacerbating, chronic kidney disease [18].

CypA levels have been examined as potential biomarkers of kidney injury. Lee et al. [19], found that elevated serum and urine CypA levels correlated with subsequent development of acute kidney injury in patients undergoing cardiac surgery. In addition, increased urine and plasma levels of CypA correlate with the progression of diabetic kidney disease [20,21], and urine CypA levels can predict microalbuminuria in children with type 1 diabetes [22]. Despite these encouraging clinical studies, the pathological role of CypA in acute kidney injury or progressive renal fibrosis has not been investigated. Therefore, the aim of this study was to determine whether CypA contributes to inflammation and kidney injury in models of acute kidney injury and of progressive renal fibrosis. To achieve this, we investigated mice lacking CypA (*CypA^−/−^*) in two disease models: acute kidney failure due to IRI, and progressive renal interstitial fibrosis following unilateral ureteric obstruction (UUO).

## 2. Results

### 2.1. CypA Deletion Protects against Acute Renal Failure, Tubular Damage and Cell Death in Renal IRI

In wild type (WT) mice, CypA mRNA levels showed a small, but significant, increase at 24 h after renal IRI (Figure 1A). Renal IRI caused an acute and severe loss of renal function in WT mice as shown by an 8-fold rise in serum creatinine levels compared to sham controls (Figure 1B). This was associated with extensive tubular damage in the inner cortex and outer medulla consisting of dilated tubules, loss of brush border, tubular cell loss/sloughing, and cast formation (Figure 1C,D). Consistent with the marked histologic damage, there was a significant increase in mRNA levels of the tubular damage marker KIM-1/HAVCR1 and a reduction in mRNA levels of the reno-protective molecule, Klotho (Figure 2A,B). A significant induction of tubular cell death was also shown by TUNEL staining (Figure 2C). 

*CypA^−/−^* mice were substantially protected from acute renal failure in the IRI model with 50% lower serum creatinine levels (Figure 1B). This protection was associated with a significant reduction in the percentage of damaged tubules (Figure 1C,D), a reduction in KIM-1/HAVCR2 mRNA levels (Figure 2A), a reduction in the number of TUNEL+ tubular cells (Figure 2C), and a significantly lesser reduction in Klotho mRNA levels (Figure 2B). 

To investigate whether CypA plays a direct role in protecting tubular cells from oxidant-induced cell death, we analysed primary cultures of tubular epithelial cells from WT and *CypA^−/−^* mice. In a dose-response study, WT and *CypA^−/−^* tubular cells showed a comparable susceptibility to H_2_O_2_-induced cell death (Figure 2D). 

### 2.2. CypA Deletion Protects against Leukocyte Infiltration in Renal IRI 

Neutrophil infiltration is a prominent response to renal IRI and plays a significant role in the induction of tubular necrosis [23,24,25]. In this study, WT mice exhibited a marked neutrophil infiltrate in the area of tubular damage (Figure 3A,B). In addition, a significant macrophage infiltrate was evident at 24 h after renal IRI, as shown by the increased CD68 mRNA levels (Figure 3C), although no significant T cell infiltrate was evident based upon CD3 mRNA levels (Figure 3D). *CypA^−/−^* mice exhibited a 73% reduction in neutrophil infiltration and a 40% reduction in CD68 mRNA levels (Figure 3A–C). 

### 2.3. CypA Deletion Does Not Protect against Tubular Damage in UUO

WT mice showed a small but significant increase in CypA mRNA levels on day 7 after UUO (Figure 4A). We also demonstrated that angiotensin II and TNF, two factors implicated in the pathogenesis of renal fibrosis in the UUO model [26], can induce CypA secretion by cultured WT tubular epithelial cells (Figure 4B). 

WT mice showed tubular dilation and atrophy, an expansion of the interstitial space and an interstitial cell infiltrate on day 7 UUO in WT mice (Figure 4C,E). Tubular damage in WT mice was also evident by the increased KIM-1 mRNA levels and reduced Klotho mRNA levels (Figure 4F,G).

*CypA^−/−^* mice showed a similar histologic pattern of tubular damage, interstitial expansion and cellular infiltration to that seen in WT mice on day 7 UUO (Figure 4D). The degree of tubular damage, shown by histology and by changes in KIM-1 and Klotho mRNA levels, was not different between *CypA^−/−^* and WT day 7 UUO (Figure 4E–G). 

### 2.4. CypA Deletion Does Not Protect against Inflammation or Fibrosis in UUO 

Infiltrating macrophages promote interstitial fibrosis in the UUO model [27,28,29,30]. A substantial macrophage infiltrate was evident on day 7 UUO in WT mice as shown by F4/80 immunostaining and CD68 mRNA levels (Figure 5A–C). The increase in both NOS2 and CD206 mRNA levels on day 7 UUO indicates the presence of both M1- and M2-type macrophage phenotypes (Figure 5D,E). In addition, immunostaining identified a significant CD3^+^ T cell infiltrate, with T cell activation indicated by increased IL-2 mRNA levels (Figure 6A,B). There was also a minor neutrophil infiltrate on day 7 UUO in WT mice (Figure 6C). 

Immunostaining showed a prominent infiltrate of F4/80+ macrophages in the day 7 UUO kidney in *CypA^−/−^* mice (Figure 5A), and this was not different compared to the WT UUO kidney (Figure 5B). An estimation of macrophage infiltration by CD68 mRNA levels showed a small, but significant, increase in *CypA^−/−^* compared to WT mice on day 7 UUO (Figure 5C). Levels of NOS2 mRNA were not different between *CypA^−/−^* and WT day 7 UUO kidney, although there was a small increase in CD206 mRNA levels in *CypA^−/−^* day 7 UUO (Figure 5D,E). T cell infiltration and activation and neutrophil infiltration were not different between *CypA^−/−^* and WT day 7 UUO kidney (Figure 6A–C). 

The day 7 WT UUO kidney exhibited marked fibrosis, as illustrated by the interstitial deposition of collagen IV and an increase in mRNA levels for collagen I and α-SMA/ACTA2 (Figure 7A–D). The pattern and degree of collagen IV deposition on day 7 UUO was not altered in *CypA^−/−^* mice (Figure 7A,B). Similarly, the increase in α-SMA mRNA levels was not different to that in the WT UUO kidney; however, there was a small, but significant, increase in collagen I mRNA levels in *CypA^−/−^* UUO (Figure 7C,D). 

## 3. Discussion

This study found CypA to be important in leukocyte accumulation and acute renal injury in the IRI model, but to be redundant in leukocyte accumulation and renal fibrosis in the UUO model. 

Necrosis is the main form of tubular cell death occurring after renal IRI, which is most prominent in the S3 segment of the proximal tubule due to its high energy requirements and disproportionate reduction in blood flow [31]. We found clear protection from IRI-induced acute renal failure and tubular cell death and damage in *CypA^−/−^* mice. This could be due to two potential mechanisms. Firstly, CypA may act directly within tubular cells to promote cell death. However, this mechanism was not supported by the finding that cultured *CypA^−/−^* tubular epithelial cells were not protected from oxidant-induced cell death. This contrasts with the significant protection against oxidant-induced cell death seen in cultured *CypD^−/−^* tubular cells and with pan-cyclophilin inhibitors such as cyclosporine [32,33]. Secondly, CypA released from damaged cells may promote the recruitment of neutrophils and macrophages to promote tubular cell necrosis and impaired kidney function. This mechanism is supported by the marked reduction in neutrophil accumulation seen in the *CypA^−/−^* mice, and by studies identifying the extracellular release of CypA during the early stages of necroptosis [34]. A role for neutrophils in promoting tubular cell necrosis in the IRI model is well-established through neutrophil depletion studies and strategies to block neutrophil recruitment [23,24,25]. The reduction in macrophage accumulation into the injured kidney in *CypA^−/−^* mice might also have contributed to protection against tubular necrosis and acute renal failure, although macrophage depletion strategies have produced variable results which are likely to be due to differences in macrophage populations depleted [35]. 

The ability of cyclosporine treatment to prevent acute renal failure in IRI models has been attributed to blockade of CypD [32,36]. However, our findings demonstrate a specific role for CypA in renal IRI. The contrast between the protection of *CypD^−/−^* tubular cells from oxidant-induced cell death [33], and our finding that *CypA^−/−^* tubular cells are not protected in the same assay, suggests that CypA and CypD contribute to acute kidney injury in renal IRI via distinct mechanisms. Our findings are consistent with studies showing that *CypA−/−* mice are protected in cardiac IRI in association with a substantial reduction in neutrophil and macrophage infiltration [37]. In addition, mice lacking CD147 are protected in renal IRI with a marked reduction in neutrophil recruitment, although this was attributed to a CD147/Selectin-E-based mechanism rather than a CD147/CypA mechanism [38]. 

Macrophages play a prominent role in the rapidly progressive renal interstitial fibrosis seen in the UUO model [27,28,29,30]. In addition, there are a small number of studies to support a role for T cells in this model [39,40,41]. In stark contrast to the findings in the IRI model, *CypA^−/−^* mice showed no reduction in the infiltration of macrophages, T cells or neutrophils. Furthermore, there was no reduction in macrophage M1 and M2 activation markers, and no reduction in T cell activation based on IL-2 mRNA levels. 

This contrast in myeloid cell accumulation between the IRI and UUO models may reflect a much greater extracellular release of CypA in the IRI model due to the substantial necroptosis. Tubular damage caused by ureter obstruction results in small numbers of apoptotic tubular cells with no evidence of necrosis and may result in little or no CypA release. We have shown that angiotensin II and TNF-α, two factors that promote macrophage infiltration and renal fibrosis in this model [26], can induce CypA secretion by tubular epithelial cells in culture. However, we were unable to measure extracellular CypA in the obstructed kidney to determine whether significant levels of CypA are released in this model. Myeloid cell infiltration in the UUO model presumably operates via different chemotactic molecules [42]. 

The lack of an effect of *CypA* gene deletion in the UUO model contrasts with studies examining gene deletion of other cyclophilin members. *CypD^−/−^* mice showed a reduction in renal fibrosis on day 12, but not on day 7 [33]. This was attributed to indirect effects of reducing tubular cell death and peritubular capillary loss [33]. In a separate study, *CypB^−/−^* mice showed a reduction in tubular dilation and macrophage accumulation on day 7 UUO, although there was no effect upon renal fibrosis based upon collagen I and *Tgfb1* mRNA levels [43]. This same study described that CypA silencing in the HK-2 tubular cell line triggers a loss of epithelial features and enhances TGF-β1-induced epithelial-mesenchymal transition [43]. However, we did not observe epithelial-mesenchymal transition (i.e., α-SMA expression by tubular epithelial cells) in *CypA^−/−^* or WT mice in the UUO model. 

Our findings in the UUO model are, in part, similar to those reported in *Cd147^−/−^* mice. Kato et al. [44], found that *Cd147^−/−^* mice were not protected from tubular dilation, tubular cell death or interstitial fibrosis on day 7 UUO, despite a reduction in macrophage infiltration. However, *Cd147^−/−^* mice did show a reduction in renal fibrosis on day 14 UUO, which was associated with a significant reduction in matrix metalloproteinase activity in the UUO kidney [44]. 

In summary, we have shown that CypA promotes neutrophil and macrophage accumulation and kidney damage in a model of acute kidney injury, but not in a model of progressive interstitial fibrosis. These findings lend weight to the concept of therapeutic inhibition of cyclophilins in the setting of acute kidney injury. 

## 4. Material and Methods

### 4.1. Animals

*CypA^+/−^* mice (129.Cg-Ppia^tm1Lubn^/J) on the 129S1/SvimJ background were purchased from JAX Mice and Services, Bar Harbor, ME, USA, and a colony maintained at the Monash Animal Research Platform. Littermate *CypA^−/−^* and *CypA^+/+^* (WT controls) mice were used. The experiments were approved by the Monash Medical Centre Animal Ethics Committee (MMCB/2015/21, 18 September 2015 to 31 December 2019) and performed according to the 8th Edition of the Australian Code of Practice for the Care and Use of Animals for Scientific Purposes.

### 4.2. Renal Ischaemia-Reperfusion Injury (IRI)

Surgery was performed as previously described [24]. Groups of 10 male mice were anaesthetized with ketamine and xylazine. Body temperature was maintained at 37 °C using a heating blanket connected to a rectal thermometer. A midline abdominal incision was made and both renal pedicles were clamped using non-traumatic vascular clamps for 19 min, during which the abdomen was temporarily sutured to minimize heat and fluid loss. Clamps were removed and reperfusion of the kidneys visually confirmed, abdominal incisions were sutured in 2 layers, and saline provided by subcutaneous injection. Analgesia was provided by subcutaneous injection of 0.05 mg/kg Buprenorphine and 4.4 mg/kg Carprofen at the end of surgery. Control animals were sham operated, having the same procedure performed, except that the renal pedicles were not clamped. 

### 4.3. Unilateral Ureteric Obstruction (UUO)

Surgery was performed as previously described [33,45]. Groups of 10 female mice were anaesthetized with ketamine and xyzaline. A midline incision was performed, the left ureter identified and ligated using two 6.0 silk sutures. The midline abdominal incision was sutured in 2 layers and analgesia provided by subcutaneous injection of 0.05 mg/kg Buprenorphine and 4.4 mg/kg Carprofen at the end of surgery. Mice were killed 7 days after UUO surgery. Control mice did not undergo surgery.

### 4.4. Renal Function 

Serum creatinine was measured using an ARL Analyzer (Dupont, North Ryde, NSW, Australia), by the Department of Clinical Biochemistry, Monash Health. 

### 4.5. Histology

Kidney histology was assessed on Periodic acid-Schiff stained 2 µm sections of formalin-fixed, paraffin embedded tissue. In the IRI model, the outer medulla was viewed at high power (400×). The percentage of tubular cross-sections exhibiting damage was scored; damage was characterized as loss of the brush border, nuclear loss, and sloughing of cells into the lumen. In the UUO model, the percentage of tubules showing dilation or atrophy were scored in the entire cortex. All analysis was performed on blinded slides. 

### 4.6. Immunohistochemistry

Immunoperoxidase staining for macrophages (rat anti-mouse F4/80; Bio-Rad, Gladesville, Australia) and collagen IV (goat anti-collagen IV; Southern Biotechnology, Birmingham, AL, USA) was performed on 4 µm sections of methylcarn-fixed tissue as previously described [33]. Immunoperoxidase staining for neutrophils (rat anti-mouse Ly6G; Abcam, Melbourne, Australia) and T cells (rat anti-mouse CD3; Bio-Rad) was performed on 5 µm cryostat sections of tissues fixed in 2% paraformaldehyde as previously described [24]. 

In the IRI model, the number of neutrophils was counted in high-power fields (400×) covering the entire inner cortex and outer medulla. In the UUO model, the number of macrophages, T cells and neutrophils was counted in high-power fields (400×) covering the entire cortex. The area of interstitial collagen IV staining in the entire cortex (excluding large vessels) was assessed under medium power (200×) by image analysis using cellSens software version 1.18 (Olympus Australia, Notting Hill, Australia).

Cell death was assessed in 4 μm sections of formalin-fixed tissue by TUNEL staining with the ApopTag Peroxidase In Situ Apoptosis Detection Kit (Millipore-Chemicon, Ryde, Australia). The number of TUNEL+ tubular cells in the inner cortex and outer medulla were counted in high-power (400×) fields. All scoring was performed on blinded slides.

### 4.7. Real Time Polymerase Chain Reaction (RT-PCR)

The total RNA was extracted from a kidney slice using the Ambion RiboPure Kit (Thermo Fisher Scientific, Scoresby, Australia) and reverse transcribed into cDNA using random primers with the SuperScript III First-Strand Synthesis System (Thermo Fisher Scientific). PCR was run on the StepOne Real-Time PCR system (Thermo Fisher Scientific) using Taqman probes. The primer/probes for α-SMA, NOS2 and CD206 have been published previously [45,46], and the other primer/probes were purchased from Thermo Fisher Scientific. The relative amount of mRNA was determined using the comparative Ct (ΔΔCt) method. All amplicons were normalized against GAPDH which was analysed in the same reaction as an internal control.

### 4.8. Cell Culture Studies

Cultures of tubular epithelial cells were prepared from normal kidneys of *CypA^−/−^* and WT mice as previously described [46]. To examine cell death, cells were starved in 1% FCS for 18 h, and then varying concentrations of H_2_O_2_ were added for 24 h. Cells then were analysed using the Cell Death Detection ELISA Kit (Roche, Mannheim, Germany) according to the manufacturer’s instructions, with results normalized to the DNA content in cell lysates using a Quant-iT DNA Assay Kit (Molecular Probes) and expressed as the ratio of optical density to DNA content. In addition, WT tubular cells were starved in 1% FCS for 18 h, and then stimulated with 10^-6^ M angiotensin II or 5 ng/mL TNF for 24 h. A CypA section in the culture media was measure by ELISA (USCN Life Science, Wangarra, Australia). 

### 4.9. Statistics

All data are shown as mean ± SD. Data were analyzed by one-way ANOVA with Tukey’s multiple comparison test, except for the analysis of CypA mRNA levels, which used the Student’s t-test. The analysis was performed using GraphPad Prism (GraphPad Prism 8.0 software, San Diego, CA, USA).

## Figures and Tables

**Figure 1 ijms-21-03667-f001:**
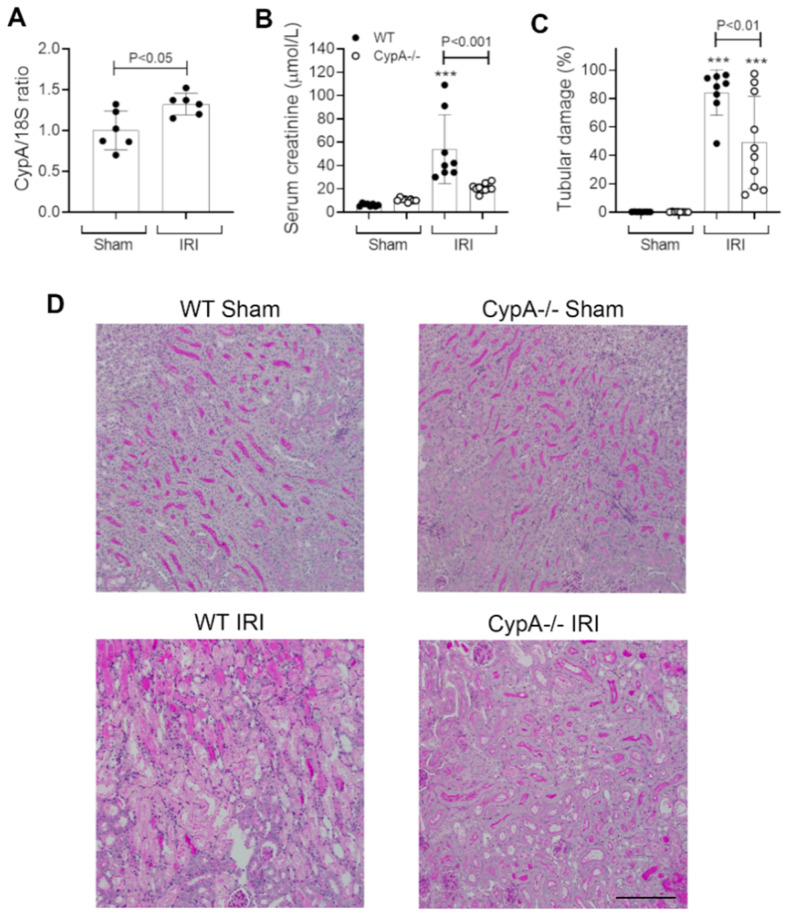
Renal function and tubular damage at 24 h in renal ischaemia/reperfusion injury (IRI) and sham controls for wild type (WT; closed circles) and *CypA^−/−^* (open circles) mice. (**A**) RT-PCR analysis of CypA mRNA levels in WT mice. (**B**) Serum creatinine levels. (**C**) Graph of tubular damage. (**D**) Periodic acid-Schiff stained kidney sections from each group. Bar = 200 μm. Data are mean ± SD. *** *p* < 0.001 versus WT sham control.

**Figure 2 ijms-21-03667-f002:**
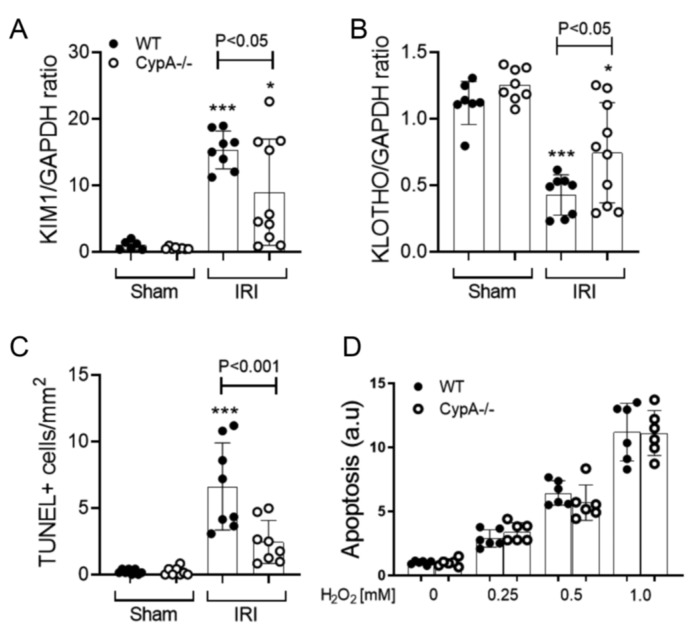
Tubular damage and cell death at 24 h in renal IRI and sham controls for WT (closed circles) and *CypA^−/−^* (open circles) mice. RT-PCR for mRNA levels of (**A**) KIM1, and (**B**) Kotho. (**C**) Quantification of the number of TUNEL+ tubular cells. (**D**) A dose-response of H_2_O_2_ induced cell death in primary cultures of tubular epithelial cells from WT and *CypA^−/−^* mice. Data are mean ± SD. * *p* < 0.05, *** *p* < 0.0001 versus WT sham control.

**Figure 3 ijms-21-03667-f003:**
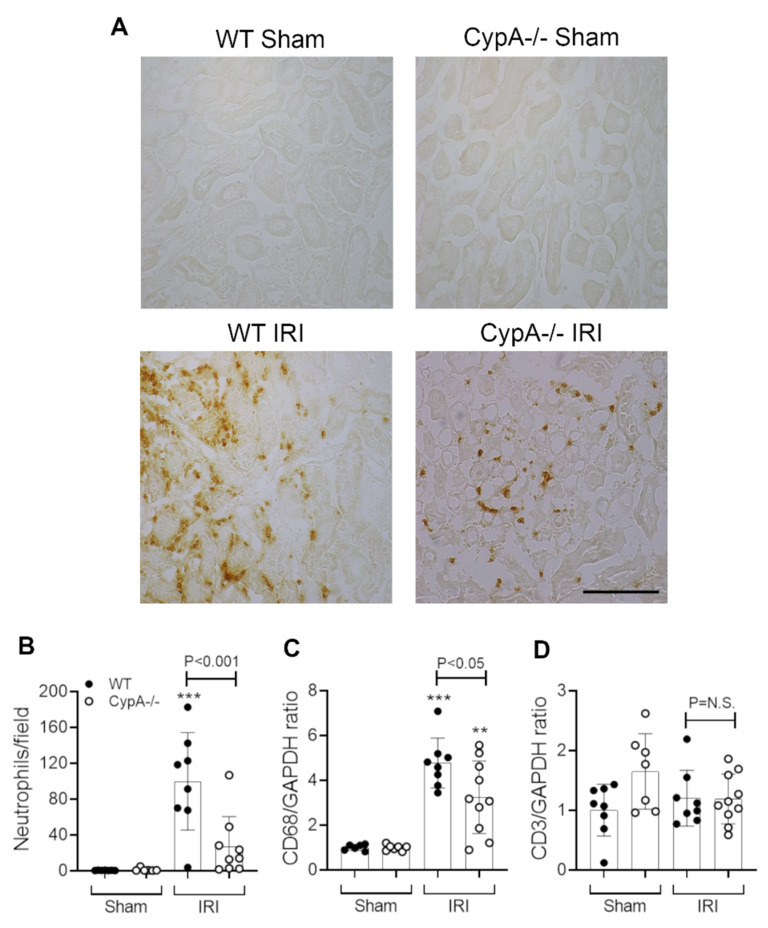
Neutrophil and macrophage infiltration at 24 h in renal IRI and sham controls for WT (closed circles) and *CypA^−/−^* (open circles) mice. (**A**) Immunoperoxidase staining for infiltrating neutrophils. Bar = 100 μm. (**B**) Quantification of neutrophil infiltration. RT-PCR analysis of (**C**) CD68, and (**D**) CD3 mRNA levels. Data are mean ± SD. ** *p* < 0.05, *** *p* < 0.0001 versus WT sham control. NS, not significant.

**Figure 4 ijms-21-03667-f004:**
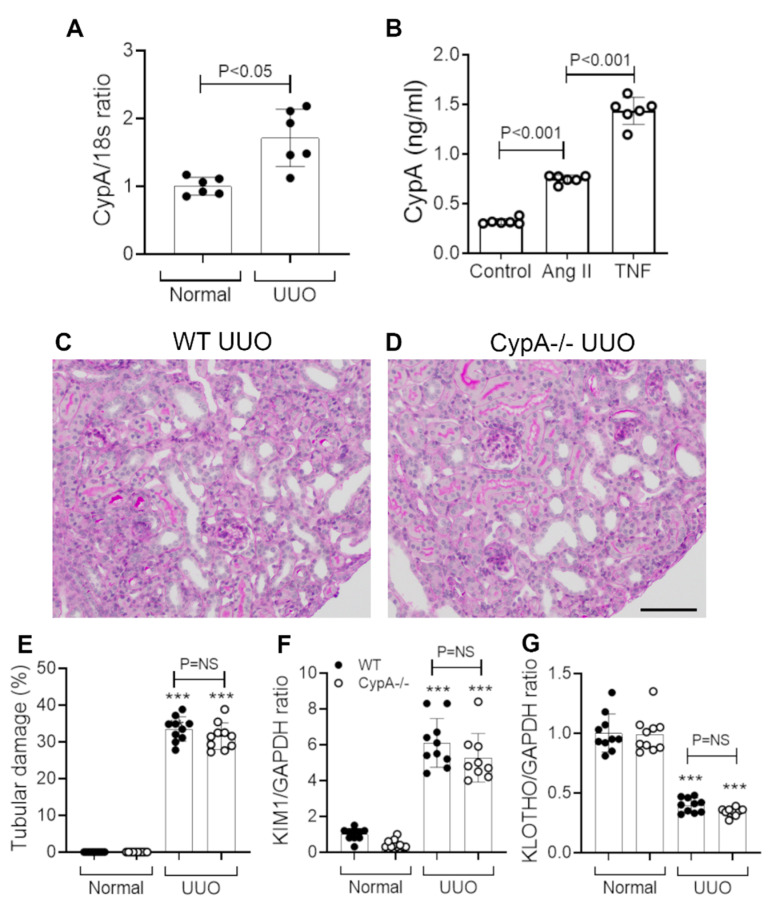
Tubular damage in day 7 unilateral ureteric obstruction (UUO) kidney compared to normal controls for WT (closed circles) and *CypA^−/−^* (open circles) mice. (**A**) RT-PCR analysis of CypA mRNA levels in WT mice. (**B**) Secretion of CypA in response to stimulation by angiotensin II or TNF in primary cultures of WT tubular epithelial cells. Periodic acid-Schiff stained kidney sections of (**C**) WT UUO and (**D**) CypA^−/−^ UUO. Bar = 100 μm. (**E**) Tubular damage score. RT-PCR for mRNA levels of: (**F**) KIM1, and (**G**) Klotho. Data are mean ± SD. *** *P* < 0.0001 versus WT sham control; NS, not significant.

**Figure 5 ijms-21-03667-f005:**
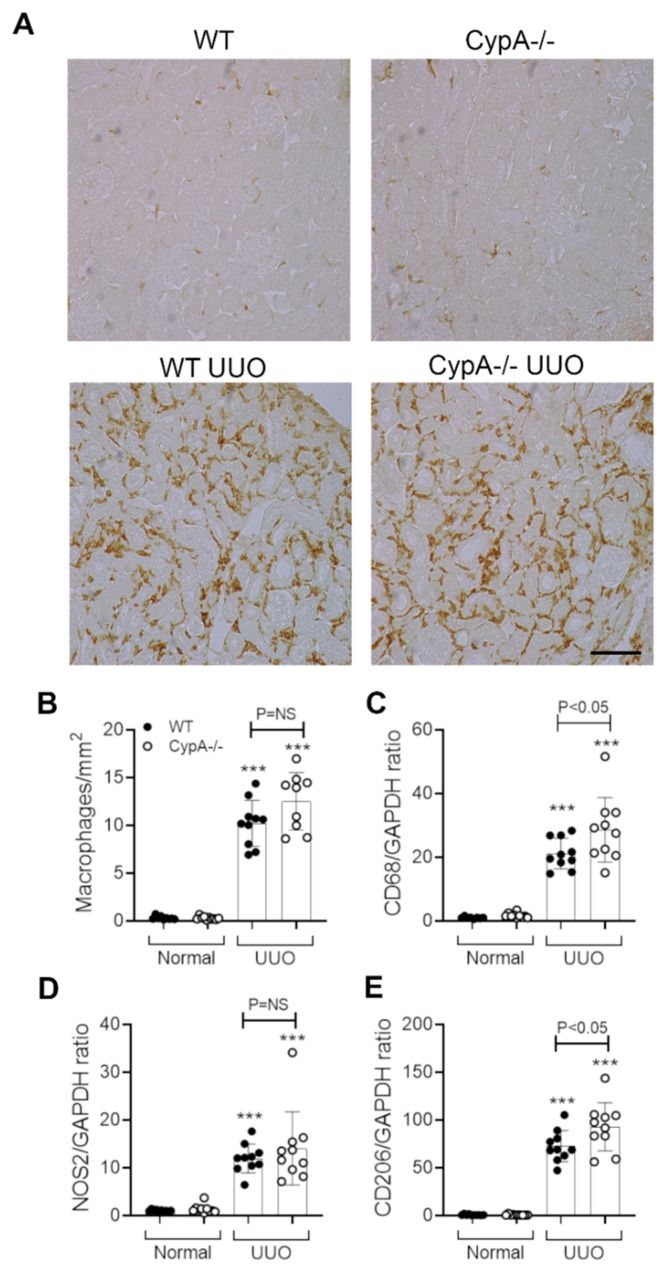
Macrophage infiltration and activation in day 7 UUO kidney compared to normal controls for WT (closed circles) and *CypA^−/−^* (open circles) mice. (**A**) Immunoperoxidase staining for F4/80^+^ macrophages in the different groups. Bar = 100 μm. (**B**) Quantification of the number of stained F4/80^+^ macrophages. RT-PCR for mRNA levels of: (**C**) CD68; (**D**) NOS2, and (**E**) CD206. Data are mean ± SD. *** *p* < 0.0001 versus WT sham control; NS, not significant.

**Figure 6 ijms-21-03667-f006:**
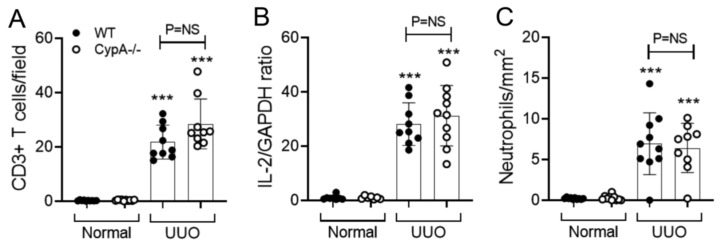
T cell and neutrophil infiltration in day 7 UUO kidney compared to normal controls for WT (closed circles) and *CypA^−/−^* (open circles) mice. (**A**) Quantification of the number of stained CD3^+^ T cells. (**B**) RT-PCR for IL-2 mRNA levels. (**C**) Quantification of the number of stained neutrophils. Data are mean ± SD. *** *p* < 0.0001 versus WT sham control; NS, not significant.

**Figure 7 ijms-21-03667-f007:**
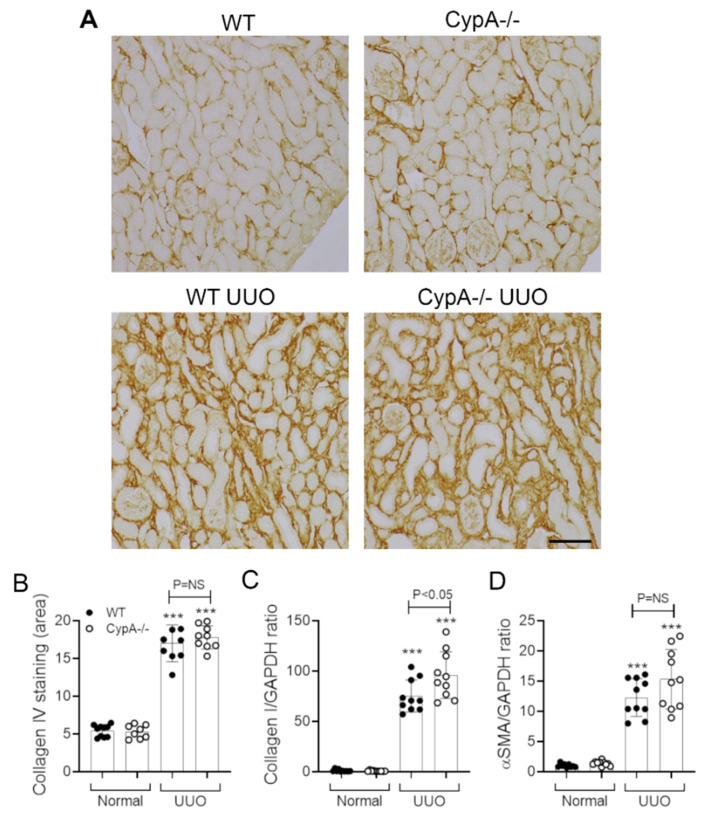
Renal fibrosis in day 7 UUO kidney compared to normal controls for WT (closed circles) and *CypA^−/−^* (open circles) mice. (**A**) Immunoperoxidase staining for collagen IV in the different groups. Bar = 100 μm. (**B**) Quantification of the area of interstitial collagen IV staining. RT-PCR for mRNA levels of: (**C**) collagen I and (**D**) α-SMA/ACTA2. Data are mean ± SD. *** *p* < 0.0001 versus WT sham control; NS, not significant.

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
