# Peer review of "Cyclophilin A Promotes Inflammation in Acute Kidney Injury but Not in Renal Fibrosis"

_ijms, 2020, doi:10.3390/ijms21103667_

Round 1
Reviewer 1 Report
In this paper Leong et al. investigated the role of Cyclophilin A (CypA) in a model of acute kidney injury (ischemia reperfusion injury – IRI) and in a model of tubulointerstitial fibrosis (unilateral ureteric obstruction –UUO). The author observed:
- CypA deficient mice are protected by IRI and in particular reduced tubular injury is evident
- CypA deficient mice have less kidney-infiltrating neutrophils after IRI at pathology analysis and have reduced level of intra-renal CD86 mRNA
- By converse UUO phenotype is not different between deficient and WT mice.
- Surprisingly some injury markers are increased after UUO in deficient mice (CD86, CD206, collagen I and SMA mRNA levels)
The authors conclude by stating that CypA promotes IRI-AKI by recruiting myeloid cells and has a neutral effect on UUO.
Please find here my observation
Data reporting in the 2 models is inconsistent. For instance there is no histology score/quantification in the UUO model (fig 4, 5 and 7). In recent years, the role of lymphoid cells in IRI/AKI has been unveiled but the authors quantify CD3 infiltration only in the UUO model.
The authors state that CypA is not toxic on tubular epithelial cells (TEC); in their in vitro experiment, deficient cells are not protected from H2O2. This is not enough to make the point. Why the authors did not use the recombinant protein to treat the cells? Moreover H2O2 is an oxidation-only model, a hypoxic chamber would have been more appropriate.
The data suggest but not prove that myeloid cell migration is driven by CypA. 1) A FACS/pathology quantification would have supported mRNA data (CD86) 2) myeloid cells could respond to enhanced tubular cells independently from CypA. Thus, blocking CypA-CD47 interaction could have proved CypA chemotactic role.
Different data in the manuscript show an evident but clinically non-significant worsening of renal injury in CypA deficient mice. However, the authors do not comment this finding. One could speculate that protein folding is relevant for regeneration/fibrosis response.
How the authors explain the difference in leucocyte recruitment between the 2 models?
Line 178-179: no data in the paper are about CypD.
Reference 26 suggests that the time-point chosen in the UUO might be inappropriate
Reference 36 finding are in contrast with the paper (about macrophages). Please reshape the sentence accordingly.
Author Response
Reviewer 1
In this paper Leong et al. investigated the role of Cyclophilin A (CypA) in a model of acute kidney injury (ischemia reperfusion injury – IRI) and in a model of tubulointerstitial fibrosis (unilateral ureteric obstruction –UUO). The author observed:
- CypA deficient mice are protected by IRI and in particular reduced tubular injury is evident
- CypA deficient mice have less kidney-infiltrating neutrophils after IRI at pathology analysis and have reduced level of intra-renal CD86 mRNA
- By converse UUO phenotype is not different between deficient and WT mice.
- Surprisingly some injury markers are increased after UUO in deficient mice (CD86, CD206, collagen I and SMA mRNA levels)
The authors conclude by stating that CypA promotes IRI-AKI by recruiting myeloid cells and has a neutral effect on UUO.
Please find here my observation
Data reporting in the 2 models is inconsistent. For instance there is no histology score/quantification in the UUO model (fig 4, 5 and 7). In recent years, the role of lymphoid cells in IRI/AKI has been unveiled but the authors quantify CD3 infiltration only in the UUO model.
Reply – we have included PCR data for CD3 in the IRI model (new Fig 3D). There is no significant T cell infiltration in this IRI model (lines 100-101). We have scored tubular damage in the UUO model based on the PAS stained sections (new Fig 4E) (lines 114, 115 and 119). No difference in tubular damage was seen between WT and CypA-/- UUO kidney, consistent with the lack of difference in mRNA levels of KIM-1 and Klotho (Fig 4F and G).
The authors state that CypA is not toxic on tubular epithelial cells (TEC); in their in vitro experiment, deficient cells are not protected from H2O2. This is not enough to make the point. Why the authors did not use the recombinant protein to treat the cells? Moreover H2O2 is an oxidation-only model, a hypoxic chamber would have been more appropriate.
Reply – this experiment was based on previous findings that CypD-/- mice are protected from tubular necrosis in the IRI model and that CypD-/- tubular cells are protected in H2O2 induced cell death. Since CypA-/- mice are also protected from tubular cell necrosis in the IRI model, we sought to determine whether CypA might have some previously unrecognised role in promoting cell death within tubular epithelial cells. Therefore, we replicated the previous study using CypD-/- tubular cells, but found that CypA-/- tubular cells were not protected. This identifies that CypA and CypD promote acute tubular necrosis via different mechanisms. While we appreciate that H2O2 stimulation mimics only the reperfusion oxidant component of IRI, this is a standard model of oxidant-induced cell death and provides a direct comparison to CypD.
The data suggest but not prove that myeloid cell migration is driven by CypA. 1) A FACS/pathology quantification would have supported mRNA data (CD86) 2) myeloid cells could respond to enhanced tubular cells independently from CypA. Thus, blocking CypA-CD47 interaction could have proved CypA chemotactic role.
Reply - we have found that CypA-/- mice with IRI have a significantly reduced accumulation of neutrophils and macrophages. We agree that this data does not point directly to an effect upon cell migration as the mechanism to explain the reduced myeloid cell accumulation. We have changed “recruitment” to “accumulation” when talking about our results (lines 167, 168, 179, 183, 211, 222), except in the graphic abstract. To prove that CypA is inducing myeloid cell accumulation via a migration-based mechanism would require intravital microscopy to visualise individual cells attaching to endothelial cells and then entering the kidney. However, use of a neutralising CD147 antibody to block such migration would not prove a role for CypA since other ligands can bind to this receptor to stimulate myeloid cells.
Different data in the manuscript show an evident but clinically non-significant worsening of renal injury in CypA deficient mice. However, the authors do not comment this finding. One could speculate that protein folding is relevant for regeneration/fibrosis response.
Reply – we consider that the very minor worsening in a minority of disease parameters does not provide grounds to speculate on a role for CypA in the UUO model.
How the authors explain the difference in leucocyte recruitment between the 2 models?
Reply – we have expanded our explanation of this point in the Discussion (lines 203-206 and 201-211).
Line 178-179: no data in the paper are about CypD.
Reply: we have re-worded this sentence to avoid this problem (lines 188-191)
Reference 26 suggests that the time-point chosen in the UUO might be inappropriate.
Reply – in ref 26, CypD-/- mice show significant protection from renal fibrosis at day 12 UUO but not at day 7 UUO. However, CypD-/- mice showed protection in a number of parameters at day 7 UUO (less tubular cell death, less peritubular capillary loss, and reduced macrophage infiltration and reduced MCP-1 and NOS2 expression), which lead to the reduced renal fibrosis seen on day 12 UUO. By contrast, CypA-/- mice at day 7 UUO showed no protection in any disease parameter, and indeed, as the reviewer points out, there was minor exacerbation of some parameters such as collagen I, CD68 and NOS2 mRNA levels. Therefore, we did not see a justification for performing an additional animal study at the day 12 UUO time point.
Reference 36 finding are in contrast with the paper (about macrophages). Please reshape the sentence accordingly.
Reply: we have re-worded this sentence to avoid this problem (lines 188-191)
Reviewer 2 Report
"In this study, the authors evaluated the role of CypA in kidney disease. Specifically, they investigated whether CypA contributes to inflammation and kidney injury in models of acute kidney injury (AKI) and of progressive renal fibrosis by evaluating mice lacking CypA (CypA-/-) in two disease models: acute kidney failure due to IRI, and progressive renal interstitial fibrosis following unilateral ureteric obstruction (UUO). They conclude that CypA promotes inflammation and AKI in renal IRI, but does not contribute to inflammation or interstitial fibrosis in a model of progressive kidney fibrosis. The authors have to be congratulated for their effort. The research design is appropriate, the methods are adequately described, the results are clearly presented and the conclusions are supported by the results. I have only some minor comments:
- A graphical abstract as a self-explanatory image to appear alongside with the text abstract in the Table of Contents should be provided.
- Although the introduction includes almost all relevant references (two recent ones should also be cited: Pediatr Diabetes. 2020 Apr 17. doi: 10.1111/pedi.13019. [Epub ahead of print] and J Immunotoxicol. 2019 Dec;16(1):182-190), it does not provide sufficient background. It should briefly place the study in a broader context to raise the interest of scientists working outside the topic of the paper, mainly clinicians. Therefore, the pathophysiology of AKI should be briefly discussed in a comprehensive paragraph highlighting the various causes (pre-renal; renal, i.e., intrinsic to the renal parenchyma; and post-renal), including the use of nephrotoxic medications such as contrast media (contrast induced nephropathy), dehydration, sepsis, renal surgery, renal ischemia, ischemia–reperfusion (IR) renal injury, and urinary tract obstruction; the clinical importance; and the complexity of the disorder. In this respect, all of the following key references can be checked among other relevant ones in the broader field of AKI research:
- Pharmacol Ther. 2017 Dec;180:99-112.
- Toxicol Rep. 2019 May 2;6:395-400.
- Biomed Rep. 2018 May;8(5):417-425.
- Food Chem Toxicol. 2017 Oct;108(Pt A):186-193.
- J Clin Med. 2020 Apr 29;9(5):E1284
I suggest that the following ones should be cited in the article:
- Pharmacol Ther. 2017 Dec;180:99-112.
- Toxicol Rep. 2019 May 2;6:395-400.
- J Clin Med. 2020 Apr 29;9(5):E1284
Author Response
Reviewer 2
Comments and Suggestions for Authors
"In this study, the authors evaluated the role of CypA in kidney disease. Specifically, they investigated whether CypA contributes to inflammation and kidney injury in models of acute kidney injury (AKI) and of progressive renal fibrosis by evaluating mice lacking CypA (CypA-/-) in two disease models: acute kidney failure due to IRI, and progressive renal interstitial fibrosis following unilateral ureteric obstruction (UUO). They conclude that CypA promotes inflammation and AKI in renal IRI, but does not contribute to inflammation or interstitial fibrosis in a model of progressive kidney fibrosis. The authors have to be congratulated for their effort. The research design is appropriate, the methods are adequately described, the results are clearly presented and the conclusions are supported by the results. I have only some minor comments:
A graphical abstract as a self-explanatory image to appear alongside with the text abstract in the Table of Contents should be provided.
Reply – we have prepared a graphical abstract of the proposed role of CypA in renal IRI. We did not include the UUO model since this was a negative result.
Although the introduction includes almost all relevant references (two recent ones should also be cited: Pediatr Diabetes. 2020 Apr 17. doi: 10.1111/pedi.13019. [Epub ahead of print] and J Immunotoxicol. 2019 Dec;16(1):182-190), it does not provide sufficient background.
Reply – we have included the J Immunotoxicol paper (lines 39-40) and the Pediatr Diabetes paper (lines 57 to 58) in the Introduction as requested.
It should briefly place the study in a broader context to raise the interest of scientists working outside the topic of the paper, mainly clinicians. Therefore, the pathophysiology of AKI should be briefly discussed in a comprehensive paragraph highlighting the various causes (pre-renal; renal, i.e., intrinsic to the renal parenchyma; and post-renal), including the use of nephrotoxic medications such as contrast media (contrast induced nephropathy), dehydration, sepsis, renal surgery, renal ischemia, ischemia–reperfusion (IR) renal injury, and urinary tract obstruction; the clinical importance; and the complexity of the disorder. In this respect, all of the following key references can be checked among other relevant ones in the broader field of AKI research:
Reply – we have included an expanded paragraph on the causes and outcomes of acute kidney injury as requested (lines 45-53).
Round 2
Reviewer 1 Report
The authors significantly improved the manuscript quality and addressed all my questions, I have no more suggestions.